# Life-Cycle Assessment and Monetary Measurements for the Carbon Footprint Reduction of Public Buildings

**Maria Rosa Trovato ***[ID]**, Francesco Nocera ***[ID]** and Salvatore Giuffrida**

Department of Civil Engineering and Architecture, University of Catania, 95125 Catania, Italy;
salvatore.giuffrida@unict.it

**\*** Correspondence: mrtrovato@dica.unict.it (M.R.T.); francesco.nocera@unict.it (F.N.);
   Tel.: +39-333-436-8621 (M.R.T.); +39-335-768-0517 (F.N.)

**Abstract:** Energy consumption in public buildings increased drastically over the last decade. Significant policy actions towards the promotion of energy efficiency in the building sector have been developed involving sustainable low-$CO_2$-emission technologies. This paper presents the results of an economic–environmental valuation of a standard energy retrofit project for a public building in a Mediterranean area, integrating a life-cycle assessment (LCA) into the traditional economic–financial evaluation pattern. The study results show that simple retrofit of sustainable low-$CO_2$-emission strategies such as wooden double-glazed windows, organic external wall insulation systems, and green roofs can reduce energy needs for heating and cooling by 58.5% and 33.4%, respectively. Furthermore, the implementation of an LCA highlights that the use of sustainable materials reduces the building's carbon footprint index by 54.1% after retrofit compared to standard materials, thus providing an additional increase in the socio-environmental–economic–financial results of 18%. Some proposals are made about the accounting of the replacement costs and the residual value as requested in the logic of life-cycle cost (that is the economic extension of the LCA), namely concerning the method to take into account the replacement costs and the residual value. The economic calculation highlights the fundamental role played by tax benefits supporting the building energy retrofit, also in temperate climate zones, thus allowing the creation of environmental benefits in addition to remarkable cost savings.

**Keywords:** envelope energy efficiency; life-cycle assessment; green public procurement; minimum environmental criteria; $CO_2$ emission reduction; cost-effectiveness analysis; financial sustainability

## 1. Introduction

Climate change has driven the necessity to reduce energy consumption and achieve low-$CO_2$-emission buildings. The European Union (EU) has committed to an 80–95% reduction of greenhouse gases (GHGs) by 2050 as part of its roadmap for moving to a competitive low-carbon economy at that date [1]. The carbon footprint refers to the total GHG emissions produced directly and indirectly by an individual, organization, event, or product expressed in the form of its carbon dioxide equivalent. The existing building stock is one of the main determinants of the carbon footprint; consequently, its energy– environmental retrofit contributes significantly to implementing the EU objective, especially if with the support of targeted tax incentives by national governments.

The integration of environmental and economic issues—thus, of common values and individual interests—especially with the prospect of an effective/efficient public fostering of the "existing building retrofit sector", needs subsidies, such as tax incentives, to be correctly determined. Nowadays, in Italy,

some fixed rates are established despite the climate zones and the specific cost-effectiveness of the different kinds of work, in some cases inconvenient from the private economic perspective. As a consequence, sometimes only few and incomplete retrofit interventions are implemented, especially in the Mediterranean climate zone and in the coastal areas, where cooling is more important than heating.

Appropriate energy–environmental–economic measurements can integrate the decision-making processes providing relevant information for (a) implementing effective/efficient policies and (b) coordinating existing building retrofit programs on the urban scale matching the objectives of the European Commission foreseen for 2050.

In this regard, since 1996 the European Commission promoted GPP (Green Public Procurement) in the Green Paper, "The public procurement in the European Union", a voluntary environmental policy tool aimed at promoting a market of low-environmental-impact products and services, based on the level of public demand. In fact, according to the estimates of the European Commission, the public annual expenditure in member countries for goods, services, and works is approximately 19% of Gross Domestic Product (GDP).

In 2003, GPP was recognized by the European Commission as a key tool of the Integrated Product Policy, in the context of the Communication COM 2003/302 and according to the European directives on public procurement (Directive 2014/23/EU, Directive 2014/24/EU, and Directive 2014/25/EU).

In Italy, the government provided a community regulatory framework (initially implemented in the Interministerial Decree of 11 April 2008, approving the national action plan on GPP), issued following the delegation granted to the government by art. 1, paragraph 1126 of Law 296/2006 (financial for the year 2007) in the Decree of the Minister of the Environment and of the Protection of the Territory and the Sea of 10 April 2013 and in the new Procurement Code (Legislative Decree no. 50/2016, amended by Legislative Decree no. 56/2017), confirming the provisions of Law 221/2015, which made Green Public Procurement (GPP) mandatory.

GPP has become also the reference in the ground of circular economy, given the implicit public authorities' commitment to more rational purchases and consumption and to the increase in environmental quality [2] of their supplies and assignments. The goals are reducing environmental impacts; protecting and improving business competitiveness; stimulating innovation; rationalizing public spending; spreading sustainable consumption and purchasing models; rationalizing the use of natural resources, in particular, energy; reducing waste; reducing the use of dangerous goods; integrating environmental issues into other corporate policies; improving the image of the public administration; and increasing the skills of public purchasers.

The mentioned "Procurement Code" has made the application of the Minimum Environmental Criteria (Law 221/2015, art. 18 and Decree 50/2016, art. 34) mandatory. Particularly, the Decree 50/2016 contains "energy and environmental sustainability criteria", i.e., the environmental requirements defined for the various stages of the purchase process, aimed at identifying the best design solution, product, or service from an environmental point of view along the life cycle. These criteria take into account also the market availability aiming at the public administration spending review.

In such a regulatory framework, the energy retrofit of public heritage [3,4] aims also to reduce $CO_2$, according to the government incentives [5], given the general economic inefficiency of the current technologies especially in the warmer area of the country [6]. These interventions can be distinguished into different subcategories to be implemented in relation to the progressive retrieval of financial resources, such as the ones involving the building envelope. Therefore, many researchers and companies have focused on methods, systems, designs, and ideas for reducing energy consumption and greenhouse gas emissions in order to achieve low-carbon buildings.

However, the building sector is characterized by different building types (e.g., residential, public, commercial, etc.) that employ a large number of technologies for heating, cooling, and lighting, and a wide variety of building envelope materials and techniques. The improvement of the building envelope is one of the retrofit programs for improving the energy performance of the built environment. The building envelope is one most significant elements in maintaining desired indoor comfort conditions

for occupants because it is the boundary with the outdoor environment and climate [7]. Therefore, it is a fundamental element—constituted of walls, windows, roofs, and floors—for the energy performance of overall building [8].

This study investigated the energy–environmental effects of the envelope retrofit [9–13], also taking into account economic–financial measurements in three stages:

1)　a technical–technological analysis of the hypothesized energy performance improvement by means of the use of sustainable materials, components, and techniques such as wooden double-glazed windows, organic external wall insulation systems, and green roofs;

2)　an environmental analysis of this project, developing the life-cycle assessment (LCA) aimed at quantifying the potential impacts on the environment and human health associated with the implementation of the above-mentioned project in order to verify its environmental effectiveness and compliance with the environmental standards [14,15]; and

3)　an economic and financial valuation aimed at identifying the cost-effectiveness and the financial sustainability of the project as a whole and in its three parts [16,17]. Due to the complementarity of economic and financial performances, such evaluations involve different areas of decision-making [18–20]. Similarly, the complexity of the environmental issue brings up practical and symbolic components [21–23] that monetary measurement can represent only partially [24–26]. Finally, as the energy retrofit programs are typically characterized by large expenses and scarcely irreversible installments and works, the correct and systematic economic assessment of these investments [27,28] supports both private and public decision making, [29–32], as well as the convergence of many axiological perspectives towards the unique ethical prospect of sustainability.

Italy is characterized by different geographical and climate zones where, according to the evolution of the costs of technological equipment, and of gas and electricity market prices, heating and cooling technology are diversely welcomed. Across the country, some technologies and works are preferred, some others rejected, despite the general environmental efficacy that makes them highly recommended by the institutions. Nonetheless, the incentive system is the same all over the country.

The results of this study aimed at providing useful guidance for envelope improvement as a tool for efficient retrofits to reduce consumption rates, empower energy efficiency, reduce $CO_2$ emissions, and rationalize spending in the public sector. Furthermore, on the basis of the breakdown of the project and of its energy reduction performance, some critical remarks and consequent proposals arose in the view of implementing a targeted incentive policy.

Section 2 presents some technological elements, a short introduction to life-cycle assessment, discounted cash flow analysis, and external benefits generated by the reduction of $CO_2$ emissions.

Section 3 introduces the case study.

Section 4 reports and discusses the results, suggesting some economic–environmental prospects that are developed as conclusions in Section 5.

## 2. Methods

According to the general aim of a fair convergence of efficiency, effectiveness and economic convenience of the building energy retrofit in the field of public real estate, we propose the implementation a standard environmental and energy retrofit program, whose economic-financial verification highlights the condition of cost-effectiveness in a temperate climate zone and some criticalities in the national energy policy based on a flat tax benefit system.

A first level of energy retrofit intervention for a public building was identified as an intervention involving the external envelope, which, representing the border between the external environment and the interior, is relevant for the internal comfort of the occupants. From the perspective of the energy efficiency of the entire building, the external envelope is characterized by walls, windows, roof, and floors.

In the following sub-section, we present the methods concerning (1) the technologies proposed to reduce the energy needs of the studied building; (2) the life-cycle assessment (LCA); (3) the software used for calculating the energy performance and the $CO_2$ emissions for entire life cycle of the elements used in the energy retrofit intervention; and (4) the economic analysis for evaluating of the financial feasibility and the cost-effectiveness of the investment.

### 2.1. Technologies

The proposed energy retrofit technologies involving the envelope of the studied building included three elements.

- *CorkPan thermal insulation.*

The self-expanded and self-glued cork panel (CorkPan) is made by a thermal roasting process, which fuses the suberin and other waxy substances contained in the cork. The completely natural roasting process does not alter the characteristics of the cork in any way but amplifies them, resulting in an increase in the volume of the single granule and an improvement of its insulating power by approximately 30%. The cork panel structure is vapor permeable, but insensitive to water and humidity, characteristics that give it exceptional dimensional stability and durability in all conditions. The extraction of the bark carried out entirely by hand, 90% of the energy for the production obtained from biomass, the unlimited life of the panel, and its total recyclability are just some details of a production process that takes place with the utmost respect for the environment and for humanity today and tomorrow.

To ensure high quality of life and constant energy savings, each intervention aimed at the overall energy efficiency of the building throughout the year, preferring materials that also guaranteed high summer phase shift values.

Cork is the most "natural" and complete choice to effectively protect the residents in the building from the summer heat and winter cold. In fact, in addition to a high thermal conductivity, which guarantees winter comfort, cork is also an excellent summer insulator, due to its high mass and thermal inertia, which can guarantee excellent values of periodic dynamic transmittance. Studies on the material show that even just a few cm of cork significantly reduces the amount of energy that flows outward through these elements.

The cork insulating coat allows increase in the internal surface temperature improving the conditions of internal comfort [33].

- *Double-glazed windows.*

Wood is the material most used for the production of windows, traditionally. It is characterized by a low coefficient of thermal transmittance, a pleasant appearance, the aptitude to be worked, and the satisfactory operating behavior of the window. The natural properties of the material are natural durability, adequate volumetric mass, dimensional stability during operation, mechanical strength and stiffness, surface hardness, and tightness of screws. The essences used were larch, in different species; spruce; pine; chestnut; Douglas fir; pitch pine; and laminated woods. Among the "latest generation" materials, wood modified with innovative procedures, such as Thermoholz and wood–plastic composites, have been introduced. The latter arise from the combination of wood and other materials in the following percentages: 50–80% wood, 50–20% polymers (Polypropylene (PP), Polyethylene (PE), Polyvinylchloride (PVC)) with the addition of additives. The wood profiles are the elements that characterize the structural parts of a fixture and, in order to guarantee the durability of the window, a correct sizing of the thickness, the average width, and their length is necessary. Wood products for passive houses generally have sections from a minimum of 70 mm up to a maximum of 100 mm, in which layers of wood alternate with layers of insulation. On the market, there are wooden fixtures with a cork core. These windows guarantee good air and water tightness, good wind resistance, excellent results in reducing external noise, and better thermal insulation. When the

window is being dismantled, however, the two elements are difficult to disassemble, not allowing the recovery of the materials used. To ensure better stability of the window frame, laminated woods are also used, obtained by gluing thin elements together so as to offset the knots. They are produced in profiles with rectangular L, Z, and T sections. The stops can have sharp or curvilinear cuts and, based on the type of stop and the profile used, the profiles can be of the simple throat or notched type. The modification of the profile morphology has led to an increase in the number of sections and an increase in the size of the leaves. Furthermore, an additional use of gaskets (two or three gaskets) and a type of open joint stop has spread. The wooden window must be protected from humidity and ultraviolet radiation through the use of paints. This latter need has produced the main innovations in the field of water-based paints, environmentally friendly and not harmful to the health of users. Among the materials used for the protection of wood from humidity, those from nanotechnologies are also spreading, which achieve optimized performance. The fixtures, made with care and precision, maintain an excellent seal over time, and require simple maintenance operations, such as painting, on average every 5–10 years.

- *Green roof.*

Green roofs are made from a layered system comprising of a waterproofing membrane, growing medium, and vegetation layer. The planted roofs mitigate solar radiation by the shading effect of plants on the soil layer and by their biological functions, such as photosynthesis, respiration, transpiration, and evaporation from soil and vegetation. Many studies that have investigated the energy performance of cool and green roofs point out their effectiveness for reducing the peak load cooling and the building cooling needs [34–36]. The energy model of the green roof is composed of a balance on the vegetation layer and soil layer [37].

The maintenance costs of these interventions were quantified on the basis of the values reported in references [38–40].

## 2.2. Life-Cycle Assessment

A life-cycle assessment technique was carried out to calculate the carbon footprint considering the GHG emissions caused by all the operational processes, transportation, and activities for the materials used in the refurbishment. A life-cycle assessment (LCA) provides the environmental impacts evaluation associated with all the stages of a product's life.

The carbon footprint was calculated with a tool called LCA calculator in terms of the following five components: production of building materials, transport of building materials, construction and demolition of buildings, direct energy use of buildings, and waste disposal. The tool evaluates all the stages of the process from the extraction and processing of raw materials, to the manufacturing, packaging and marketing processes, to the use, re-use, and maintenance of the product, and on to its eventual recycling or disposal as waste.

The calculation of the carbon footprint involves the analysis of the products' life cycle, including both direct and indirect carbon emissions of the related activities, and comparing the results with other carbon emission studies. The methodology for determining the carbon footprint has been studied for several years [41] and it is mainly concentrated on energy consumption and carbon emissions associated with production of building materials, building construction process, or a single building [42].

LCA calculator was based on ecoinvent database which builds on the method of life-cycle assessment (LCA) as standardized by International Organization for Standardization (ISO) 2006. The basic building blocks of the ecoinvent database are Life Cycle Inventory (LCI) datasets, representing the individual unit processes of human activities and their exchanges with the environment.

The embodied carbon in materials and material production processes were calculated according to ISO 14020 and ISO 14040, as well as ISO 14025. The embodied $CO_2$ includes energy consumption of building materials and products, the use of raw materials, and greenhouse gases. In the calculations, the greenhouse gases were transformed into their $CO_2$ equivalent ($CO_2$ eq) by using IPCC's characteristic

factors, in which the corresponding factors for $CH_4$ and $N_2O$ are 25 and 298, respectively. The sum of these was the $CO_2$ emissions from embodied energy of building materials and products [43].

### 2.3. Simulation Software

Energy simulation software "Design Builder" was used to model a typical public building in a Mediterranean climate and to evaluate its performance and improvement potential through envelope retrofit strategies. Design Builder, which provides a graphical interface for the numerical code Energy Plus [44], was used to evaluate the energy needs (Q) and primary energy (PE) for space heating and cooling. In this study, Design Builder simulated the building energy performance under various retrofit scenarios.

The model was created and calibrated using the data available in the energy audit report. In a second phase, each retrofit intervention was considered to estimate the yearly energy benefits.

Once the potential annual primary energy saving of each simulated retrofit option was determined, a carbon footprint analysis was then carried out.

### 2.4. Economic Analysis: Monetary and Contents

According to the two general aims of the economic–financial verification—that is financial feasibility and economic profitability—in the energy–environmental sustainability aspect, the methodology followed was two stages.

(1) At the first stage, the costs of the works and the revenues coming from government incentives and energy savings were calculated according to project–economic items and environmental–economic items.

a. Project–economic items

The costs of the works envisaged by the retrofit program were calculated by associating the parametric prices of typical works as listed in the Bill of Quantities of the Public Works currently in force in Sicilian Region. The works also included the related temporary structures, such as scaffolding and the removal of the hypothetical old plaster and window frames, typical in standard buildings. The costs for the design and supervision were calculated as a percentage of the costs of the works; the annual costs [45] for maintenance were calculated as differentiated percentages for each type of work and accounted for along its life cycle.

Revenues included the energy savings at current unit prices of electricity, and the government incentives calculated as a percentage of the work and divided into ten constant annuities from the first year.

b. Environmental–economic items

The environmental–economic issues were a further component of the revenues concerning the monetary value of the carbon footprint reduction measured as below explained (Section 2.5). Such a valuation provided the environmental external benefit, whose economic and financial results were considered separately and compared to the simple project–economic results.

The external environmental benefits monetary measurement was calculated by multiplying the positive balance of carbon footprint in terms of emissions of tonnes of $CO_2$ equivalent, by the most prudent value indicated in the literature as explained below. Namely, two approaches have been developed measuring the benefit due to the reduction of the environmental cost in terms of $CO_2$ emissions, the first one known as social cost of carbon (SCC), the second one called marginal abatement cost (MAC).

(2) At the second stage, a discounted cash flow analysis (DCFA) [46,47] was carried out in order to calculate the cost-effectiveness and the financial sustainability of the project.

The methodological pattern of life-cycle cost, the replacement cost, and the residual value of the works were accounted in DCFA performed over a 100-years lifespan within which the energy–environmental performance should be ensured:

$$C_G = C_I + (C_M + C_O) \left[ \frac{q^T - 1}{rq^T} \right] \mp V_R \left( \frac{1}{q^T} \right) \tag{1}$$

where $C_G$ global cost; $C_I$ investment costs; $C_M$ maintenance costs; $C_O$ operating costs; $V_R$ residual value; $r$ discount rate; $q = (1 + \mathrm{r})$; $T$ lifetime span.

According to the principle of the persistence of the current conditions concerning costs; market prices; regulations and incentives, such as the tax benefits in force in Italy, at the end of the lifetime of each work, the incomings and outgoings were accounted again according to above-mentioned conditions.

The cost-effectiveness and the financial sustainability of the project were calculated from the perspective of different results and indices.

(1) The net present value (*NPV*), that is the sum of the incoming and outgoing cash flows, that is revenues (*R*) and costs (*C*), over a defined time horizon (*T*), discounted at the discount rate *r*. NPV is less, equal, or more than the (net) future value (FV) if the discount rate (*r*) [48,49] is more, equal, or less than 0; *NPV* is expected to be significantly positive in case of a private player:

$$NPV = \sum_{i=0}^{T} \frac{R_i - C_i}{(1 + r)^i} \geq 0 \tag{2}$$

(2) The total rate of return (*TRR*), that is the more significant index of profitability thus the ratio between *NPV* and the present cost; *TRR* should be greater than the opportunity cost of capital $c_k$.

$$TRR = \frac{\sum_{i=0}^{T} \frac{R_i - C_i}{(1+r)^i}}{\sum_{i=0}^{T} \frac{C_i}{(1+r)^i}} \geq c_k \tag{3}$$

(3) The internal rate of return (*IRR*), that is the discount rate $r_{IRR}$ at which $NPV = 0$, that is the maximum rate of return that can be extracted by an investment; it only depends on the distribution of the stream along the time horizon of the investment:

$$\sum_{i=0}^{T} \frac{R_i - C_i}{(1 + r_{IRR})^i}) = 0 \tag{4}$$

(4) The external rate of return (ERR)—also called modified internal rate of return (MIRR)—refers to both the cost of the investment and the interests on reinvested cash, and is calculated on the basis of an interest rate external to the investment at which net (positive) cash flows generated by the investment over its time horizon can be invested or borrowed $r^*$. The external rate of return $r_e^*$ is the rate at which the investment costs discounted at the rate $r$ equals the future value at time $T$ of the positive cash flows ($CF_{i(>0)}$) deferred at the rate $r^*$, given $CF_i = R_i - C_i$. In other words, ERR is the IRR of an ideal investment whose unique cost is the initial investment cost calculated as the NPV at the rate $r$ of the negative cash flows over the time horizon $T$, and whose unique revenue is the future value (at year $T$) of the positive cash flows at the rate $r^*$. This particular *IRR* is $r_e^*$.

$$\sum_{i=0}^{T} \frac{CF_{i(<0)}}{(1 + r_e^*)^i} = \sum_{i=0}^{T} CF_{T-1(>0)} (1 + r^*)^{T-i} \tag{5}$$

(5) The elasticity ($E_r$), that is the marginal *NPV* at the discount rate *r*:

$$E_r = \frac{\frac{\delta NPV_r}{NPV_r}}{\frac{\delta r}{r_r}} \tag{6}$$

(6) The discounted payback period (*DPP*) is the number of years it takes to break even from undertaking the investment cost ($I_0$) by discounting future cash flows and recognizing the time value of money ($r > 0$) [50]; the higher the discount rate, the longer the *DPP*. More simply, a payback period (*PP*) can be calculated without taking into account the time preference rate ($r = 0$) [51]. In general, *PP* is the ratio between the total investment cost and the annual constant or average cash flow. Often, the variability of the cash flow over the lifetime of the project reduces the reliability of the formulas usually implemented for *DPP*, so that a more general formula can be proposed considering $NPV(i)$, and then:

$$DPP = i_{NPV(i)=0} \tag{7}$$

(7) The average period at the rate $r$ ($P_r$) [52], that is a sort of time elasticity, that can be considered as the average period of deferment of the $i_{th}$ annual net discounted cash flows ($CF_i$) given the discount factor;

$$P_r = \frac{\sum_{i=0}^{T} \frac{iCF_i}{(1+r)^i}}{\sum_{i=0}^{T} \frac{CF_i}{(1+r)^i}} \tag{8}$$

The discount rate $r$ is an important indicator of the intertemporal solidarity practiced with the implementation of the project, and it enables two different and complementary prospects, the private one as means, the public one as end.

Concerning the first one, the discount rate can be assumed as the well-known weighted average cost of capital (*WACC*), referred to the funds in terms of debt (*D*) and equity (*E*);

$$WACC = \frac{i_d D + i_e E}{D + E} \tag{9}$$

where $i_d$ is the interest rate for debt and $i_e$ is the opportunity cost of equity that can be respectively referred to the active and passive interest rates charged to households and consumers, according to the statistics of Bank Italia (2019), set at 4.66% (over 5 years loan life) and 0.12%; assuming a leverage of 50%, *WACC* is 2.39%.

Concerning the second one, that has the function and consequently, the size of the social discount rate, the extensive literature on the subject agrees on the need to keep this at the lowest possible levels, therefore, compatible with the pillars of the sustainability of the relationship between the social system and the environment.

Since the retrofit project supposes different actions—thermal insulation, windows and green roof—the above-mentioned indices were calculated for both the whole retrofit program and each action.

*2.5. Integrating Externalities into the DCFA*

Once integrated into the DCFA, the valuation of the environmental externalities from the building energy retrofit provides a cost–benefit analysis (CBA) [53,54]. The socio-economic externalities could be the external costs from the $CO_2$ emissions, or the external benefit from the reduction of $CO_2$ emissions.

The calculation of the monetary value of a tonne of $CO_2$, i.e., the externalities produced by the energy retrofit considered, could be carried out on the basis of two different approaches:

- The first approach is the social cost of carbon (SCC).

The social cost of carbon (SCC) is a metric commonly employed to evaluate the expected economic damages from $CO_2$ emissions; it is a monetary measure of the long-term damage done by the $CO_2$ emissions in a given year. This measurement also represents the monetary value of the damages avoided/created for the emission reduction/increase, and makes it possible to estimate the damage caused by climate change, i.e., in terms of losses of net agricultural productivity, impacts on human health, increases in vulnerability of the territory exposed to flood risk, and increases of energy needs for buildings' heating and cooling. Currently, in relation to the modeling used and due to the limited

data availability, the measurement of the SCC cannot include all kinds of damage, such as the physical, ecological, and economic impacts due to climate change as highlighted in the literature, and confirmed in the fifth IPCC assessment report [55]. However, the current SCC estimates are a useful measure for assessing the climatic impacts of changes in $CO_2$ emissions. Estimates of the social cost of carbon vary because of different assumptions about future emissions, how climate will respond, the impacts this will cause, and the way we value future damages [56,57].

To calculate SCC, the Interagency Working Group (IWG) pools the outputs from three different integrated assessment models (IAMs) [58].

DICE (Dynamic Integrated Climate–Economy model) was developed by William Nordhaus (Yale University) [59]; FUND (Framework for Uncertainty, Negotiation, and Distribution model) was originally developed by Richard Tol (University of Sussex); and PAGE (Policy Analysis of the Greenhouse Effect model) was developed by Chris Hope (University of Cambridge). The evaluations of the social cost of carbon using the three different integrated assessment models (IAMs) reported for 2020 based on a discount rate [60] of 3%, a value of SCC of $74 per tonne of $CO_2$ by implementing of PAGE09, $40 per tonne of $CO_2$ by implementing of DICE-2010R, and $22 per tonne of $CO_2$ by implementing of FUND3.8.

A survey of 23 richer OECD nations [61] and the European Union found wide variations in the approach to and level of carbon valuations. For policy appraisal, it found countries including Chile, Canada, the USA, France, the UK, and Germany using an average 2014 price of $56/tonne of $CO_2$, rising to $115 in 2050.

New estimates of SCC that were produced by Ricke K. et al. [62] report an average value of $417.74 per tonne of $CO_2$ with 66% confidence intervals of $177–805 per tonne of $CO_2$.

- The second approach is called marginal abatement cost (MAC).

A marginal abatement cost (MAC) curve is defined as a graph that indicates the cost, associated with the last unit (the marginal cost) of emission abatement for varying amounts of emission reduction (in general in million/billion tonnes of $CO_2$). Therefore, a baseline with no $CO_2$ constraint must be defined in order to assess the marginal abatement cost against this baseline development. A MAC curve allows one to analyze the cost of the last abated unit of $CO_2$ for a defined abatement level while obtaining insights into the total abatement costs through the integral of the abatement cost curve. The average abatement costs can be calculated by dividing the total abatement cost by the amount of abated emissions. According to the underlying methodology, MAC curves can be divided into expert-based and model-derived curves.

Expert-based MAC curves assess the cost and reduction potential of each single abatement measure (including new technologies, fuel switches, and efficiency improvements) based on educated opinions. Subsequently, the measures are explicitly ranked from cheapest to most expensive to represent the costs of achieving incremental levels of emissions reduction.

Another widespread approach to MAC curves is to derive the cost and potential for emission mitigation from energy models. The most common way is to distinguish models into economy-oriented top-down models and engineering-oriented bottom-up models. In both cases, abatement curves are generated by summarizing the $CO_2$ price resulting from runs with different strict emission limits or by summarizing the emissions levels resulting from different $CO_2$ prices. Bottom-up energy models are partial equilibrium models representing only the energy sector in contrast to top-down models, which cover endogenous economic responses in the whole economy. Bottom-up models are either simulation models or optimization models that calculate a partial equilibrium either through the minimization of the system costs or by maximizing consumer and producer surplus [63].

In the European Union, a MAC approach has been used as it is reflected in the European Union Emission Trading System (EU ETS). Nevertheless, different values have been included in cost–benefit analysis, which are different from the current price in the EU ETS which fixed a price of about €40/tonnes $CO_2$-eq until 2020 [64], which is, however, quite close to the average of the SCC values calculated by

the IWG pools through the three different integrated assessment models and for a social discount rate of 3.3% (the value provided in the guidelines for cost–benefit analysis for Italy) [65]. Based on the data collected, we used €40/tonnes of $CO_2$-eq as the price of carbon emissions.

## 3. Materials

The reference building investigated was a public building located in the old town of Modica (Ragusa) (latitude 36°51′31″ N and longitude 14°45′39.23″ E), see Figure 1. This area was characterized by 121 days in winter for a heating period and 122 days for the cooling period. The building, erected in 1962, had a rectangular plan and included two floors, for a total floor area of 473 m².

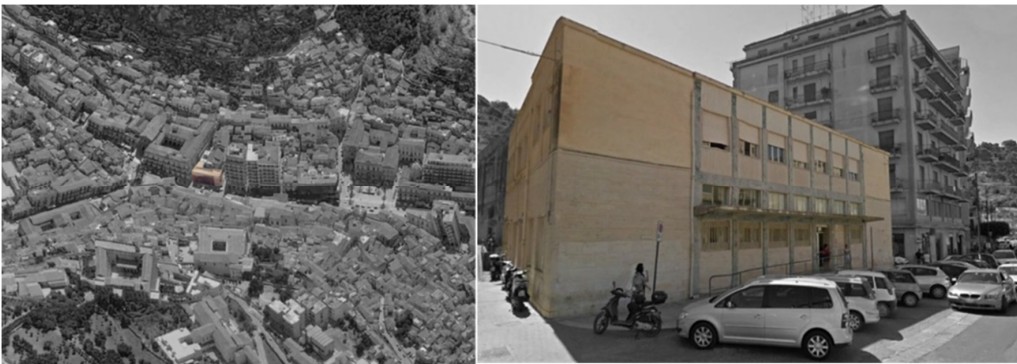

**Figure 1.** Former Post Building.

The building was made of reinforced concrete and brick walls, without any insulating material, and concrete roof and ceilings. The main geometric features and the thermo-physical characteristics of the building components are reported in Table 1. Table 2 describes the features of the construction elements and the openings.

**Table 1.** Thermo-physical features of envelope components before the retrofit interventions.

| Geometric Features | Envelope Component | U-Value (W·m$^{-2}$·K$^{-1}$) | Superficial Mass (kg·m$^{-2}$) |
|---|---|---|---|
| Heated gross volume (V) = 2000 m³ | External walls | 1.477 | 606.13 |
| Total external surface (S) = 911 m² | Flat roof | 1.592 | 395.50 |
| Shape factor (S/V) = 0.455 m$^{-1}$ | Ground floor | 0.934 | 1131.50 |
| Net floor area (Sn) = 473 m² | Windows | 5.850 | - |

**Table 2.** Thermo-physical features of envelope components before the retrofit interventions.

| External Walls | Thickness (m) | Thermal Conductivity (W·m$^{-1}$·K$^{-1}$) | Density (kg·m²) | Thermal Capacity (J/kg$^{-1}$·K$^{-1}$) |
|---|---|---|---|---|
| Marble cladding | 0.03 | 3.00 | 2800 | 1000 |
| Cement mortar | 0.03 | 1.35 | 2000 | 1000 |
| Limestone blocks | 0.2 | 2.00 | 2000 | 1000 |
| Air gap | 0.1 | 0.66 | 1.3 | 1008 |
| Perforated bricks | 0.08 | 0.40 | 775 | 840 |
| Lime/gypsum plaster | 0.02 | 0.80 | 1300 | 1000 |
| **Roof** | **Thickness (m)** | **Thermal Conductivity (W·m$^{-1}$·K$^{-1}$)** | **Density (kg·m²)** | **Thermal Capacity (J/kg$^{-1}$·K$^{-1}$)** |
| Gravel | 0.04 | 1.2 | 1700 | 1000 |
| Bituminous waterproofing membrane | 0.005 | 0.23 | 1100 | 1000 |
| Mortar | 0.06 | 1.35 | 2000 | 1000 |
| Load-bearing floor-slab | 0.2 | 0.6 | 918 | 840 |
| Lime/gypsum plaster | 0.02 | 0.8 | 1300 | 1000 |
| **Openings** | **Thickness (m)** | **Thermal Conductivity (W·m$^{-1}$·K$^{-1}$)** | **Ug-Value (W·m$^{-2}$·K$^{-1}$)** | **Uf-Value (W·m$^{-2}$·K$^{-1}$)** |
| Clear glass | 0.003 | 0.90 | 5.89 | - |
| Aluminum | 0.002 | 230 | - | 3.00 |

As previously stated, the study was carried out using Design Builder, which provides the graphical interface for the Energy Plus code. In this way, it was possible to carry out accurate thermal analyses and allow very detailed inputs, containing climatic data (including air temperature, solar radiation, relative humidity hourly profiles); construction materials and components in dedicated libraries or manually edited; energy systems' specifications; and time schedules (systems' management, occupancy, electric lighting, ventilation, etc.). All the rooms in the building were considered occupied zones equipped with an air conditioning system for both heating and cooling. The primary energy consumption was calculated using an energy efficiency ratio (EER) of 3.20, neglecting seasonal variations. The air conditioning system was set to operate at a temperature of 19 °C during the heating period (1 December–31 March) and 26 °C during the cooling period (1 June–30 September). Internal gains were taken into account by considering an occupancy density of 0.05 people·m$^{-2}$ and a lighting and equipment power density of 4.5 W·m$^{-2}$. An air change rate of 0.5 vol·h$^{-1}$ was used for air quality purposes.

The building envelope was characterized by opaque vertical closures made by limestone blocks and perforated bricks with an internal air gap.

All the external walls were coated by marble cladding applied on blocks of limestone with a cement mortar layer.

The stratigraphy of building walls was based on similar buildings set up in the same period in Sicily.

The building flat roof was not viable and was covered by a gravel layer to save the waterproofing membrane from solar radiation.

## 4. Application, Results, and Discussions

### 4.1. Energy Needs at the Current State

The yearly energy consumption of the building, due to a very low thermal resistance of the envelope (both opaque and transparent), was significantly high.

The overall value of specific energy demand was 47.1 kWh·m$^{-2}$·year$^{-1}$ for heating and 96.34 kWh·m$^{-2}$·year$^{-1}$ for cooling. It is interesting to point out that the cooling energy needs were higher than the heating energy needs.

### 4.2. Building Retrofit

Starting from the results obtained through the building energy simulations and according to the consumptions related to different services, the following typologies of intervention were proposed:

(1)   improvement of building envelope performance by increasing the thermal insulation;
(2)   replacement of the windows; and
(3)   installation of a green roof.

More precisely, the proposed solution was based on the replacement of marble cladding with a 6 cm panel of Corkpan (insulation cork board) ($\lambda = 0.036$ W·m$^{-1}$·K$^{-1}$) on the outer side of the external walls, as well as the addition of a 2 cm thick gypsum plaster on the outer side of the external masonries.

An extensive green roof was considered because of its low additional load, i.e., it does not require any additional strengthening, and consequently, it is particularly suitable for existing building structures. The vegetation types used were mosses, sedum, graminaceous, and succulent plants (leaf area index 5.0 m$^{-2}$·m$^{-2}$) that are very common and suitable plants to be used on an extensive green roof in a Mediterranean climate [66]. They are small plants that grow across the ground rather than upwards, offering good coverage and roof membrane protection [67]. The substrate was a thin layer (10 cm) of porous soil; it was typically a mixture of sand, clay, mineral aggregates, and organic matter. The soil is above the filter layer, a geotextile fabric, which filters the soil granules in order to prevent the filling of the drainage layer [68].

The intervention on the external windows consisted of the replacement of the current glazing with a double-glazing (s = 4 mm) separated by an air gap (s = 18 mm) compatible with the wooden frame. The glass had the further following characteristics: low-emissivity coating on the inner glazing ($\varepsilon$ = 0.1) and selective coating on the outer glazing, which allowed for a solar gain factor g = 0.5. The values of the thermo-physical properties and the thermal transmittance of the envelope components after the retrofit intervention are reported in Table 3. Here, it is also reported, in brackets, the rate of improvement of the thermal transmittance as well as the superficial mass after the retrofit interventions. It can be observed a sensible reduction of U-value for all components, especially for the external walls (−73.6%), as well as a remarkable increment of thermal mass, especially for the green roof (+66.2%).

**Table 3.** Thermo-physical features of envelope components after the retrofit interventions.

| Flat Roof Components | Thickness (m) | Thermal Conductivity ($W \cdot m^{-1} \cdot K^{-1}$) | Density ($kg \cdot m^2$) | Thermal Capacity ($J/kg^{-1} \cdot K^{-1}$) |
|---|---|---|---|---|
| Ground | 0.15 | 1.00 | 1500 | 2000 |
| Geotextile fabric | 0.0005 | 0.22 | 1800 | 910 |
| Expanded clay | 0.1 | 0.92 | 900 | 1000 |
| Anti-roots barrier | 0.0005 | 0.25 | 1200 | 1800 |
| Bituminous waterproofing membrane | 0.005 | 0.23 | 1100 | 1000 |
| Mortar | 0.06 | 1.35 | 2000 | 1000 |
| Loadbearing floor slab | 0.2 | 0.6 | 918 | 840 |
| Lime/gypsum plaster | 0.02 | 0.8 | 1300 | 1000 |
| U-value ($W \cdot m^{-2} \cdot K^{-1}$) | 1.154 (−27.5%) | | | |
| Superficial mass ($kg \cdot m^{-2}$) | 657.5 (+66.2%) | | | |
| **External Walls Components** | **Thickness (m)** | **Thermal Conductivity ($W \cdot m^{-1} \cdot K^{-1}$)** | **Density ($kg \cdot m^2$)** | **Thermal Capacity ($J/kg^{-1} \cdot K^{-1}$)** |
| Lime/gypsum plaster | 0.02 | 0.8 | 1300 | 1000 |
| Corkpan™ | 0.06 | 0.036 | 130 | 1900 |
| Limestone blocks | 0.2 | 2.00 | 2000 | 1000 |
| Air gap | 0.1 | 0.66 | 1.3 | 1008 |
| Perforated Bricks | 0.08 | 0.40 | 775 | 840 |
| Lime /gypsum plaster | 0.02 | 0.80 | 1300 | 1000 |
| U-value ($W \cdot m^{-2} \cdot K^{-1}$) | 0.39 (73.6%) | | | |
| Superficial mass ($kg \cdot m^{-2}$) | 470.58 (−22.4%) | | | |
| **Wooden Window** | **Thickness (m)** | **Thermal Conductivity ($W \cdot m^{-1} \cdot K^{-1}$)** | **$U_g$-Value ($W \cdot m^{-2} \cdot K^{-1}$)** | **$U_f$ -Value ($W \cdot m^{-2} \cdot K^{-1}$)** |
| Low emission double glass 4/18/4 | 0.036 | 0.90 | 1.65 | - |
| Wood | 0.1 | 0.12 | - | 0.98 |
| U-value ($W \cdot m^{-2} \cdot K^{-1}$) | 1.65 (−71.8%) | | | |

After the retrofit, the calculations of energy needs were carried out under the same hypothesis in terms of occupancy, artificial lighting, and equipment [69,70] and an air change rate of 0.3 vol·h$^{-1}$.

The results of the energy need for heating and cooling are reported in Figure 2. The specific energy demand for heating became 19.51 kWh·m$^{-2}$·year$^{-1}$ (see Figure 2), while for cooling became 62.85 kWh·m$^{-2}$·year$^{-1}$. The energy savings achieved through the proposed solutions reached a reduction of 58.50% for heating and 33.40% for cooling. This result was mainly due to the reduction in the transmission heat losses, especially in the roof.

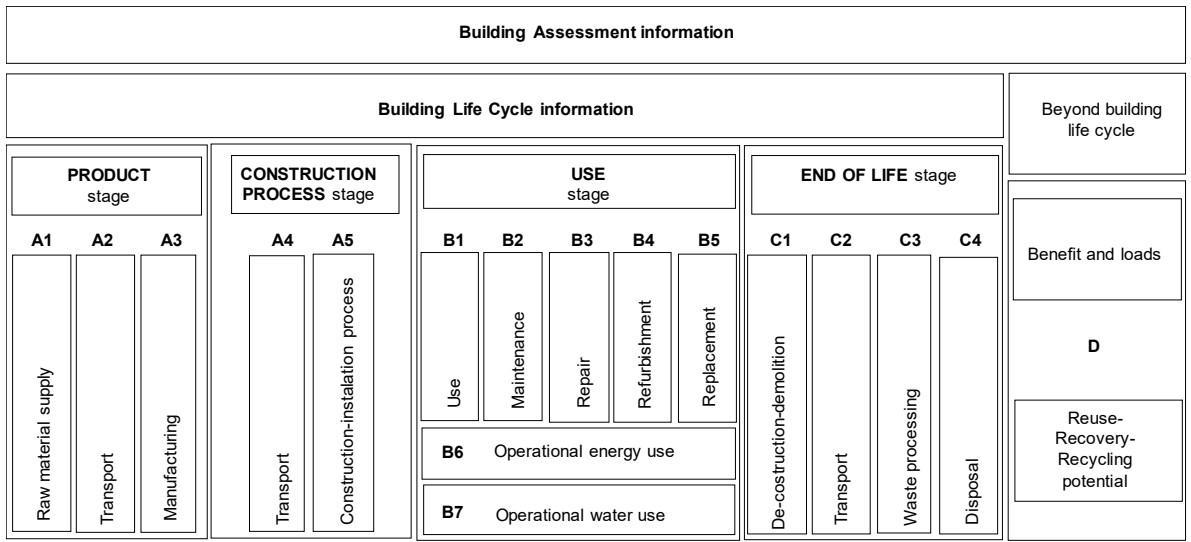

**Figure 2.** Life-cycle stages from BS EN 15978:2011.

### 4.3. LCA Imaging

The assessment of the system boundary for this study is summarized in Figure 2 in accordance with EN 15978 Sustainability of Construction Works Assessment of Environmental Performance of Buildings and it was carried out based on the life-cycle assessment (LCA) method.

The carbon footprints for the materials were calculated by using the unit impact value (in kg $CO_2$ eq/kg of product) obtained from the ecoinvent database.

Table 4 reports the building's life steps and carbon footprint for a building life of 100 years calculated by the LCA tool before the retrofit intervention. The carbon footprint associated with the building construction process corresponded to 1952.7 tonnes of $CO_2$ eq [71]. This value includes the assembly in a construction site (86.5 tonnes of $CO_2$ eq) and disassembly of materials, components, and structures at life end (107 tonnes of $CO_2$ eq). In accordance with the literature, the most significant effect on carbon footprint was due to the use of reinforced concrete structures (204 tonnes of $CO_2$ eq) and brick (36 tonnes of $CO_2$ eq) [72]. Considering a 100-year lifespan, the construction process affected 30% of the carbon footprint and the maintenance process affected 4% of the carbon footprint, while the use of the building affected 66% of the carbon footprint [73].

**Table 4.** Building's carbon footprint for a building life of 100 years.

| Building's Life-Cycle Steps | $CO_2$ Equivalent ($CO_2$ eq) Emissions (kg·$CO_2$ eq) |
| --- | --- |
| Site activities, transportation, and ground use | 86,500 |
| Screed and foundation | 46,800 |
| Structure | 54,800 |
| Loadbearing floor slab | 83,800 |
| Exterior walls | 46,000 |
| Internal works | 27,400 |
| Roof | 50,000 |
| Windows | 7400 |
| Electric and thermal system, waterworks | 10,400 |
| Maintenance | 76,600 |
| Electricity for lighting, cooling, heating, and other uses | 1,153,000 |
| Demolition and afterlife treatment | 107,000 |
| Net $CO_2$ equivalent emission | 1,952,700 |

For the retrofit and improvement of the energy performance of building envelope, simple and widely used technologies and systems were adopted.

Above all, organic external wall insulation systems (insulation cork board), a green roof [74–80], and wooden double-glazed windows were chosen for their sustainable low $CO_2$ emissions [81]. Insulation cork board (ICB), also known as black expanded cork board, was used as an external wall insulation system for the retrofit of building due to its excellent insulation characteristics [82].

In addition, it is a renewable material, made with low-value cork or forestry residues, obtained from the periodic pruning of cork trees, by removing the bark from the cork tree's branches [83]. According to the ISO/TS 14067(2013) requirements [84], Table 5 reports the Corkpan's life cycle had a negative value of net $CO_2$ emissions of 116.229 kg $CO_2$ eq/m$^3$.

**Table 5.** Corkpan's life-cycle steps [7].

| Corkpan's Life-Cycle Steps | Carbon Footprint (kg $CO_2$ eq/m$^3$) |
|---|---|
| $CO_2$ embodied in Corkpan | −272.186 |
| Extraction from tree | 145.843 |
| Transportation | 8.628 |
| Production | 1.485 |
| Net $CO_2$ equivalent emissions | −116.229 |

In Table 6, the green roof's life cycle is reported.

**Table 6.** Green roof's life-cycle steps.

| Green roof's Life-Cycle Steps | Carbon Footprint (kg $CO_2$ eq/m$^2$) |
|---|---|
| $CO_2$ embodied in green roof | 18.50 |
| Transportation | 5.90 |
| Usage | 0.12 |
| Carbon sequestration | −0.62 |
| Net $CO_2$ equivalent emissions | 23.90 |

Several researchers agree that thermal performance and carbon footprints for wooden window frames are better than PVC and aluminum frames [85,86]. The carbon footprint of an aluminum window frame is almost four times as high as that of the wooden window frame, while the carbon footprint of a PVC window frame is twice as high as a wooden window frame [87]. After the retrofit, the calculations of carbon footprints were carried out they are reported in Figure 3.

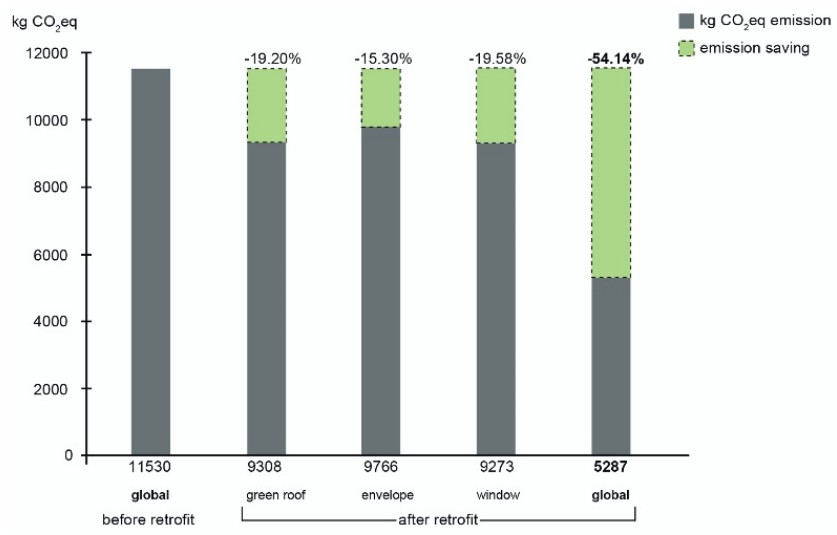

**Figure 3.** Carbon footprint before and after retrofit.

The carbon footprint became 5287 kg $CO_2$ eq/year (see Figure 3); consequently, the proposed solutions reached a reduction of 54.14%. This value was only due to a reduction of electrical consumptions and derived emissions after the interventions of the retrofit. Moreover, Figure 3 reports the reduction of $CO_2$ eq of each retrofit intervention, in which it is noticeable the replacement of windows offered major effectiveness [88–90].

### 4.4. Economic Analysis and Valuation Issues

As explained above, the economic analysis was carried out turning the relevant items of the retrofit project into monetary measurements whose economic–financial significance has been represented by means of the synoptic report of results and indices.

As first, the retrofit program's economic items with the related prices were listed as the input of the database, based on which the calculation model was created (Table 7).

Accordingly, the single amounts were grouped and summed up, distinguishing stock and streams, and indicating the time distribution of the latter, for example, 10 years for incentives, and the entire lifetime of the project as for savings, maintenance costs, and so on (Table 8).

**Table 7.** List of the works for the implementation of the three energy-environmental retrofit actions.

| Actions | Id. | Description | U.m. or % | Unit Price (€/U.m.) | Extent | Total Amount (€) |
|---|---|---|---|---|---|---|
| Termal insulation | 1 | Scaffolding 30 days | m² | 8.52 | 631.8 | 5383 |
| | 2 | Scaffolding over 30 days | m² | 0.24 | 631.8 | 150 |
| | 3 | Corkpan works | m² | 0.50 | 631.8 | 316 |
| | 4 | Corkpan plaster demolition | m² | 1.50 | 631.8 | 948 |
| | 5 | Corkpan materials | m² | 23.60 | 631.8 | 14,909 |
| | 6 | Corkpan construction | m² | 40.00 | 631.8 | 25,270 |
| | 7 | Wall painting | m² | 13.20 | 631.8 | 8339 |
| | 8 | Delivery to dump | m³ | 24.70 | 19.0 | 468 |
| | 9 | Dump fees | m³ | 36.00 | 19.0 | 682 |
| | 10 | Additional expenses | 7% | | | 8471 |
| | 11 | Incentives | 90% | | | 76,242 |
| | 12 | Savings | m² | 1.21 | 631.8 | 768 |
| | 13 | Maintenance | 1% | | | 149 |
| Windows | 14 | Reflective windows | m² | 42.00 | 92.4 | 3881 |
| | 15 | Thermal break windows | m² | 489.40 | 92.4 | 45,225 |
| | 16 | Windows removal | m² | 14.20 | 92.4 | 1312 |
| | 17 | Doors removal | m² | 14.20 | 3.0 | 43 |
| | 18 | New doors | m² | 544.40 | 3.0 | 1633 |
| | 19 | Dump fees | m³ | 12.00 | 3.8 | 46 |
| | 20 | Additional expenses | 7% | | | 4483 |
| | 21 | Incentives | 65% | | | 41,626 |
| | 22 | Savings | m² | 30.65 | 95.4 | 2924 |
| | 23 | Maintenance | 0,5% | | | 28 |
| Green roof | 24 | Green roof | m² | 50.00 | 242.0 | 12,100 |
| | 25 | Additional expenses | 15% | | | 1210 |
| | 26 | Incentives | 50% | | | 6050 |
| | 27 | Savings | m² | 12.89 | 242.0 | 3118 |
| | 28 | Maintenance | 3,15% | | | 121 |

**Table 8.** Incoming and outgoing stock (€) and streams (€/y).

| Stocks/Streams | Overall | Thermal Insulation | Windows | Green Roof |
|---|---|---|---|---|
| Investment cost—building works (stock) | −130,123 | −60,418 | −55,790 | −13,915 |
| Maintenance (annuities over the 30-y lifetime) | −585 | −149 | −55 | −381 |
| Incentives (annuities over 10 y) | 9076 | 5082 | 3389 | 605 |
| Savings (annuities over the 30-y lifetime) | 5060 | 824 | 1667 | 2569 |
| External environmental benefit (annuities over the 30-y lifetime) | 250 | 71 | 90 | 89 |

Then, based on the appraisal of the overall and detailed incomings and outgoings, a discounted cash flow analysis was carried out (Table 9) in order to measure the overall and detailed (by each of the three actions to be implemented) cost-effectiveness and financial sustainability of the project. The graph of Figure 4 shows the temporal structure of this investment for the relationship between the investment cost and the time distribution of the cash flows; the investment was significantly characterized by the 10-year tax incentives without which the project would not have been cost-effective at all.

**Table 9.** Discounted cash flow analysis (DCFA) report: excerpt of the discounted overall and detailed cash flows over 40 years of the projects, 100 years lifetime at 2.39% discount rate (€).

| Year | Overall | Thermal Insulation | Windows | Green Roof |
|------|---------|--------------------|---------|------------|
| 0 | 130,123 | 60,418 | 55,790 | 13,915 |
| 1 | 13,234 | 5622 | 4884 | 2728 |
| 2 | 12,925 | 5491 | 4770 | 2664 |
| 3 | 12,624 | 5363 | 4659 | 2602 |
| 4 | 12,329 | 5237 | 4550 | 2541 |
| 5 | 12,041 | 5115 | 4444 | 2482 |
| 6 | 11,760 | 4996 | 4340 | 2424 |
| 7 | 11,485 | 4879 | 4239 | 2368 |
| 8 | 11,217 | 4765 | 4140 | 2312 |
| 9 | 10,956 | 4654 | 4043 | 2258 |
| 10 | 10,700 | 4545 | 3949 | 2206 |
| 11 | 1469 | 520 | 1243 | −294 |
| 12 | 1435 | 508 | 1214 | −287 |
| 13 | 1401 | 496 | 1186 | −280 |
| 14 | 1369 | 485 | 1158 | −274 |
| 15 | 1337 | 473 | 1131 | −267 |
| 16 | 1305 | 462 | 1104 | −261 |
| 17 | 1275 | 451 | 1079 | −255 |
| 18 | 1245 | 441 | 1053 | −249 |
| 19 | 1216 | 431 | 1029 | −243 |
| 20 | 1188 | 421 | 1005 | −238 |
| 21 | 1160 | 411 | 981 | −232 |
| 22 | 1133 | 401 | 959 | −227 |
| 23 | 1107 | 392 | 936 | −221 |
| 24 | 1081 | 383 | 914 | −216 |
| 25 | 1055 | 374 | 893 | −211 |
| 26 | 55,632 | 365 | 55,790 | −206 |
| 27 | 5039 | 356 | 4884 | −201 |
| 28 | 4921 | 348 | 4770 | −197 |
| 29 | 4807 | 340 | 4659 | −192 |
| 30 | 4694 | 332 | 4550 | −188 |
| 31 | −9147 | 324 | 4444 | 13,915 |
| 32 | 7385 | 317 | 4340 | 2728 |
| 33 | 7212 | 309 | 4239 | 2664 |
| 34 | 7044 | 302 | 4140 | 2602 |
| 35 | 6880 | 295 | 4043 | 2541 |
| 36 | 6719 | 288 | 3949 | 2482 |
| 37 | 3949 | 281 | 1243 | 2424 |
| 38 | 3856 | 275 | 1214 | 2368 |
| 39 | 3766 | 269 | 1186 | 2312 |
| 40 | 3678 | 262 | 1158 | 2258 |

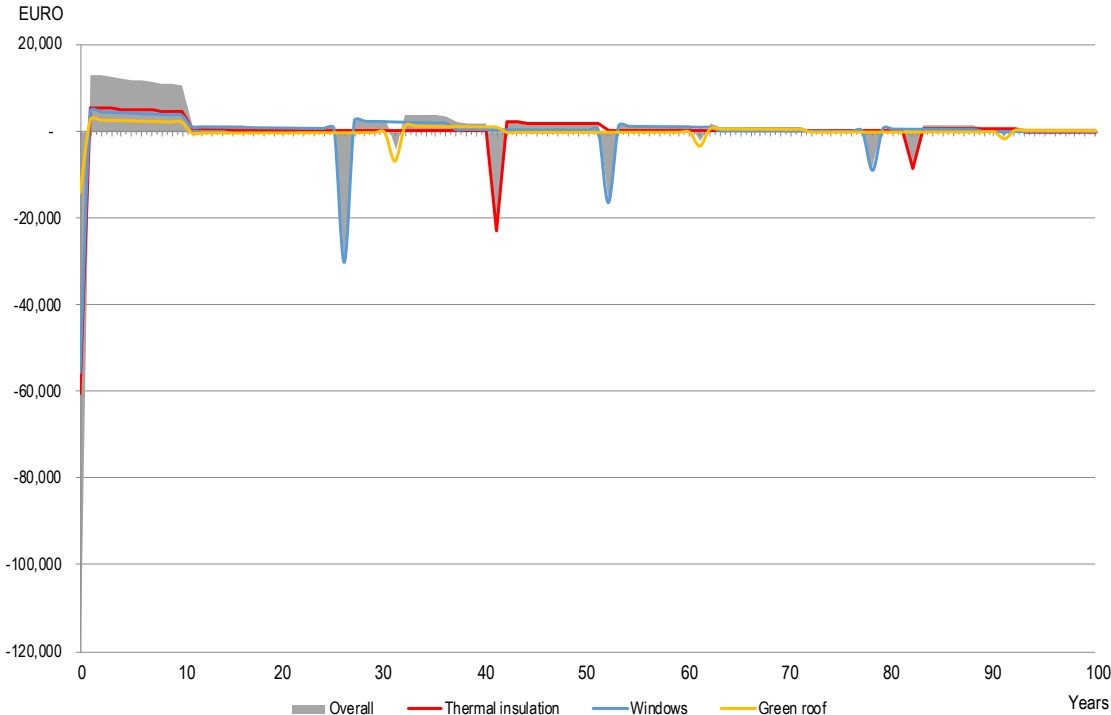

**Figure 4.** Display of the discounted cash flow detailed by items (thermal insulation, windows, and green roof).

Finally, the two cost-effectiveness calculations (excluding and including the environmental external benefits calculated as the monetary measurement of the carbon footprint reduction) were performed, as reported and compared in Table 10.

**Table 10.** Overall and detailed economic results and financial indices.

| Indices | Overall | | Thermal Insulation | | Windows | | Green Roof | |
|---|---|---|---|---|---|---|---|---|
| | Non Env Benefit | with Env. Benefit | Non Env Benefit | with Env. Benefit | Non Env Benefit | with Env. Benefit | Non Env Benefit | with Env. Benefit |
| *NPV* | 12,973 | 20,368 | −1332 | 1305 | 4424 | 7756 | 9881 | 11,308 |
| *TRR* | 8.52% | 13.37% | −2.02% | 1.97% | 7.64% | 13.40% | 34.84% | 39.87% |
| *IRR* | 3.15% | 3.58% | 2.19% | 2.58% | 2.88% | 3.24% | 12.94% | 13.94% |
| *ERR* | 2.45% | 2.48% | 2.38% | 2.40% | 2.43% | 2.46% | 2.64% | 2.67% |
| *E* | 3.60 | 2.55 | 11.47 | 13.22 | 5.46 | 3.43 | 0.73 | 0.72 |
| *DPP* | 21 | 19 | 42 | 35 | 23 | 21 | 6 | 6 |
| *P* | 75 | 72 | 87 | 80 | 80 | 76 | 48 | 48 |

These results show that within the general cost-effectiveness of the project—aiming, as a whole, at improving the building from the thermal wellness perspective—some specific points need to be distinguished.

The first concerns the economic insignificance of thermal insulation, due to its low energy performance in such a warm climate zone; in addition to the negative and/or low economic results and financial indices, the discount payback period (*DPP*) and the average period (*P*) of this action were greater than the project lifetime and insignificant, both with and without the contribution of the monetary value of the carbon footprint reduction.

The second is the significance of elasticity, a very important economic–financial characteristic in the case of a project whose specific environmental target matches different cultural levels, financial situations, and civil commitments of householders, companies, and public administrations. In addition

to the percentage value *E*, elasticity can be represented as the *NPV/r* function; Figure 5 displays and compares the *NPV/r* function of the project and those of the single actions: for each, the overall elasticity is represented by the average slope of the curve, while the cost-effectiveness is represented by its position. The graph shows that the overall curve crosses all the others, thus having the greatest *E* and the most variable *NPV*. The less elasticity of the other curves and their different position suggests the need for a global strategy able to coordinate such heterogeneous environmental and economic-financial values of a retrofit project.

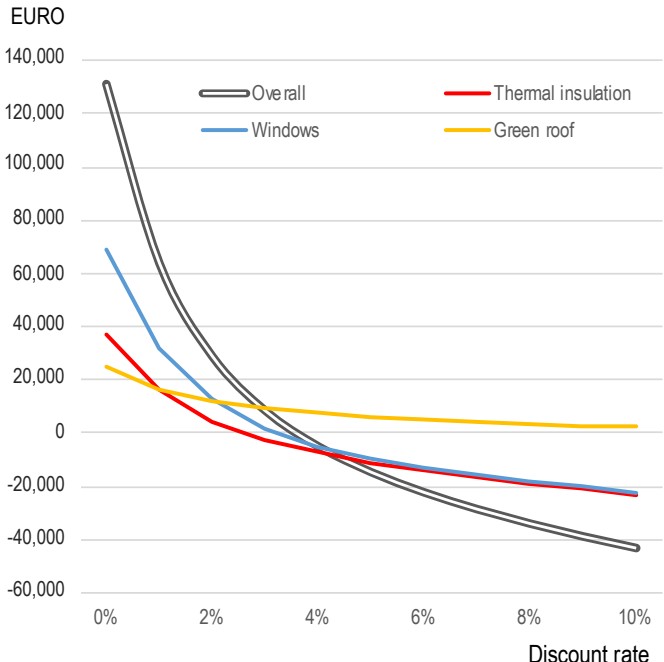

**Figure 5.** Elasticity of the overall and partial actions of the project including the account of the monetary value of the carbon footprint reduction (*r* in the x-axis; *NPV* in the y-axis).

The third concerns the discounted payback period (*DPP*) of the project, both in its overall extension and as for its parts. As previously observed, the overall and partial extents of *DPP* are coherent with the other economic and financial results. An insight, concerning the display of the compared cumulated cash flows in the 30-years project lifetime (Figure 6), confirms that the *DPP* strongly depends on the incentives. Particularly concerning the cumulated cash flow function of thermal insulation, we observed that if the incentives were prolonged three more years, the *DPP* would be 14 years instead of >30; also in this case it is obvious that the lower the discount rate, the shorter the *DPP*. In a zero discount rate hypothesis, it would be: $DPP_{Overall} = 9\ y$; $DPP_{Thermal\ insulation} = 25\ y$; $DPP_{Windows} = 10\ y$; $DPP_{WinGreen\ roofdows} = 3\ y$.

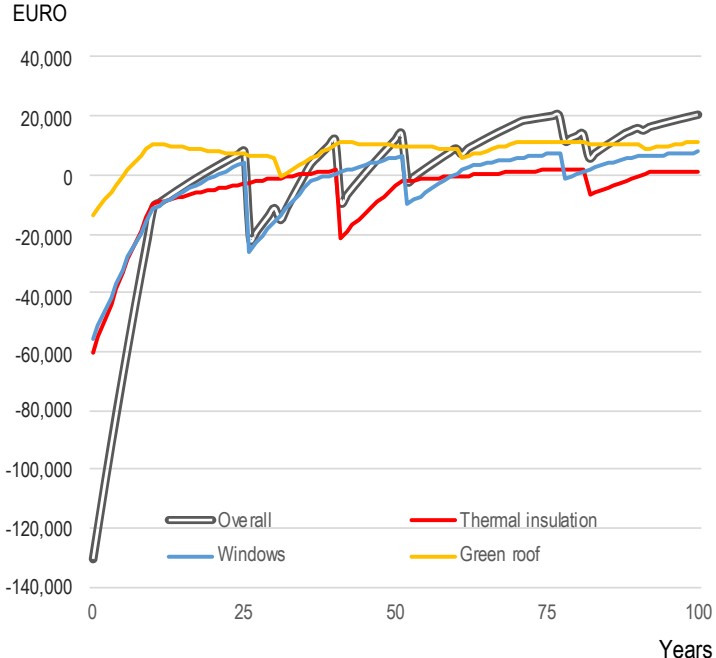

**Figure 6.** Cumulated discounted cash flows of the overall and partial actions of the project (x-axis time; y-axis cumulated discounted cash flow).

A fourth point that is connected to the previous ones is the temporal measurement of elasticity, that is the average period ($P_r$). This particular index is not usually taken into account in financial analysis for two main reasons: on the one hand, it is very sensitive to the variability of the cash flow; on the other hand, it can be considered a measurement of the willingness to differ. Due to the specific time dimension of $P$, although it is a financial index, a greater willingness to differ typically characterizes economic players by their specifically environmental perspective and generally sustainability-oriented perspective.

## 5. Conclusions

### 5.1. Energy–Environmental Issues

Thermal performance and carbon footprints of three common retrofit interventions were evaluated to provide an important approach towards designing sustainable buildings.

The strategies for improving the building energy efficiency are well-known but the importance of $CO_2$ equivalent emissions of building materials is often ignored.

This paper highlighted how the $CO_2$ equivalent emissions from embodied energy was an important share and could be fundamental in the suitable strategies of retrofit interventions. The results highlighted the significant effects of each proposed constructive solution in terms of $CO_2$ eq reduction (18%).

The approach of considering not only the energy but also the carbon footprint in the analysis of a building retrofit can be a useful strategy that can lead to the reduction of air pollution emissions and improved wellbeing.

Life-cycle assessment (LCA) made it possible to quantify the impacts on the environment and human health associated with the three sub-categories of intervention into which the retrofit of the external envelope of the building was divided, verifying the performance in environmental terms, or rather the congruence of the latter of the Minimum Environmental Criteria, within the framework of the GPP, as made mandatory by Legislative Decree 56/2017 "Procurement Code".

From a technical–technological and environmental point of view, the solutions could reduce the energy demand for heating (58.5%) and cooling (34.4%) in addition to lowering the carbon footprint of the building (54.14%).

*5.2. Economic–Environmental Issues*

Concerning the economic and financial perspective, the following can be proposed on the completion of this experience.

First, in continuity with the thermo-technical issues, we observed that the climate zone, inversing the seasonal typical energy efficiency of the usual retrofit processes, affected the cost-effectiveness of the project, that in this case resulted in higher costs in the summer period thus reducing the economic performance of the investment.

Second, the decisive role of the incentives needs to be noted for their progressive adaptation to the different climate zones and to the different actions composing the overall energy–environmental retrofit project.

Third, and according to the previous two points, the relationship between "economic rationality" and "environmental effectiveness" should be taken into account for the support that DCFA provides to the decision-making process performed in the energy–environmental industry.

The previous topics can be developed starting from some considerations made in a previous experiment [91], concerning the behavioral pattern connecting individual axiology (private interest) and public ethics (common values) toward the joint objective of sustainability. In that experiment, concerning a more complex and articulated project divided into nine actions, the different performances levels of the actions allowed us to select and group them in several and differently cost-effective bundles, the results of which are worth considering for the overarching objective of sustainability.

In this regard, also according to the considerations on the average period previously calculated, the interpretation of this index proposed above inversed the mainstream perspective of the valuation of investments, so that further remarks can be made:

- according to the traditional perspective, the shorter $P$, the more cost-effective the investment; while, according to the inverse and complementary perspective, the longer $P$, the less risky the project;
- since the latter statement can be assumed only if the cost-effectiveness of the project is significant, "project" can be considered a dimension of economic acting that differs from the "investment" in terms of vision; as such, project can be considered complementary to investment for the following reasons:

  ○ the "investment" is considered economically profitable and financially sustainable based on monetary measurements: the greater the latter, the greater the cost-effectiveness of the investment; no limit must be imposed on profitability;
  ○ the "project" instead underlies a wider decision-making context, in which the financial soundness compensates for lower profitability, especially in case of investment involving social, cultural, environmental, territorial, urban, or landscape capital.

According to these remarks, some proposals can be made. First, incentives should be adequately targeted according to climate zone and type of works and installments, in order to encourage also the implementation of less profitable interventions having, however, high environmental efficiency. Second, a progressive incentive, linked to the completeness of the intervention, could be introduced; in this case, the in-depth analysis of the thermo-technical, energy–environmental, and economic calculation plays a strategic role. Lastly, instead of incentivizing the individual works separately, a single and progressive incentive rate could be established for the entire amount of the works based on the degree of energy autonomy achieved by the building as a whole and not separately for the individual real estate units.

These hypotheses suggest further progress of energy–environmental policy [92], which could concern the coordination of large territorial entities, such as denser building fabrics, where further incentives could be given for projects of environmental energy retrofits that generate positive impacts on the scale of urban heat islands (UHI). This expression defines indeed territorial entities having an energy–environmental delimitation which introduces a further theme in the field of urban studies, the one of energy performance in terms of primary energy requirements and consequent carbon footprint.

**Author Contributions:** Conceptualization, M.R.T., F.N., and S.G.; methodology, M.R.T., F.N., and S.G.; software, M.R.T., F.N., and S.G.; validation, M.R.T., F.N., and S.G.; formal analysis, M.R.T. and S.G.; investigation, M.R.T. and F.N.; resources, M.R.T. and F.N.; data curation, M.R.T., F.N., and S.G.; writing—original draft preparation, M.R.T., F.N., and S.G.; writing—review and editing, M.R.T. and S.G.; visualization, M.R.T., S.G., and F.N.; supervision, M.R.T. and S.G.; project administration, M.R.T., F.N., and S.G. Specifically: M.R.T. and S.G. All authors have read and agreed to the published version of the manuscript.

**Funding:** This research received no external funding.

**Conflicts of Interest:** The authors declare no conflict of interest.

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
