# Peer review of "Life-Cycle Assessment and Monetary Measurements for the Carbon Footprint Reduction of Public Buildings"

_sustainability, doi:10.3390/su12083460_

Round 1

Reviewer 1 Report

The review results determine that the following will be a better paper if the following contents are supplemented.

1) Modifying the topic and keywords

Title; Life-Cycle Assessment and monetary measurements for the carbon footprint reduction of public buildings → Life-Cycle Assessment for the carbon footprint reduction of public buildings

Keywords: envelope energy efficiency, Life Cycle Assessment, Green Public Procurement,  Minimum Environmental Criteria, CO2 emission reduction, cost-effectiveness analysis; financial sustainability→envelope energy efficiency, Life Cycle Assessment, Minimum Environmental Criteria, CO2 emission reduction, cost-effectiveness analysis( not more than five)

2) in chapter two

2.2 Life Cycle Assessment

2.4. Economic analysis:

2.5. Integrating external sources into the DCFA

→Remove 2.5 and include it in 2.2 and 2.3

3)  in chapter four

4.3 LCA imaging

4.4. Economic analysis and Valuation Issues

→Combine 4.3 and 4.4 to 4.3(LCA)

4) in Chapter 5

Chapter 5 also comprehensively covered three topics of energy, environment and economy, which need to be reduced.

 In other words, focus on life cycle assessments for reducing carbon emissions in public buildings. On the other hand, I would like to reduce the economic and environmental contents.

Therefore, if the above is supplemented, your paper is for the Carbon Footprint Reduction of Public Buildings and is expected to be a good topic and good content.

Author Response

We would like to thank the reviewers and Editor for their detailed comments and suggestions for the manuscript.

We believe that the comments have identified important areas which required further analyses and improvement.

After completion of the suggested edits, the revised manuscript has benefitted from an improvement in the overall presentation and clarity.

Below, you will find a point by point description of how each comment was addressed in the manuscript. Original reviewer comments in black color and responses in red color.

Reviewer 1

The review results determine that the following will be a better paper if the following contents are supplemented.

Comments 1:

1) Modifying the topic and keywords

Title; Life-Cycle Assessment and monetary measurements for the carbon footprint reduction of public buildings → Life-Cycle Assessment for the carbon footprint reduction of public buildings

Keywords: envelope energy efficiency, Life Cycle Assessment, Green Public Procurement, Minimum Environmental Criteria, CO2 emission reduction, cost-effectiveness analysis; financial sustainability→envelope energy efficiency, Life Cycle Assessment, Minimum Environmental Criteria, CO2 emission reduction, cost-effectiveness analysis (not more than five).

Answer to the comment 1

This work consolidates the collaboration between the area of ​​environmental technical physics within which the LCA develops and the area of ​​the science of assessments within which the DCFA takes place.

The DCFA provides an economic-monetary indication (profitability for the promoter) to support private choices on investments that have public relevance, given that in this case the private entity performs an almost total exclusive public service

Having to take into account this effective collaboration, in view of a more solid behavioural pattern in the context of economic planning of energy retrofit interventions, we believe that the environmental component is to be taken into consideration precisely in the context of the overlap and complementarity between private interests and values shared, which can be detected by the science of evaluations.

Therefore, we believe that the correct title is the one originally set, namely: "Life-Cycle Assessment and monetary measurements for the carbon footprint reduction of public buildings", as that suggested by the review, or "Life-Cycle Assessment for the carbon footprint reduction of public buildings” would fail to grasp the actual nature of the issue addressed.

As for the auditor's suggestion on reducing the number of keywords, we have done so.

Comments 2:

2) in chapter two

2.2 Life Cycle Assessment

2.4. Economic analysis: (comment 2.a)

2.5. Integrating external sources into the DCFA (comment 2.a)

→Remove 2.5 and include it in 2.2 and 2.3 (comment 2.b)

Answer to the comment 2a

We present in sub-section 2.5 which in the old version of the paper had the title "Integrating external costs into the DCFA" and which in the empty version has the title 2.5. Integrating externalities into the DCFA” a literature review that highlights the main approaches used for the evaluation of the externalities produced by CO2 emissions. The determination of the CO2 value is instrumental to proceed with the DCFA.

In this regard, this sub-section is present in the paper following sub-section 2.4. which in the old version had the title "Economic analysis" and which in the new version became "Economic analysis: monetary and contents", in which we present the methodological approach for financial feasibility and economic profitability (DCFA).

In fact, the choice of how to determine the value of the CO2 emissions that is related to one aspect of the DCFA, comes after the presentation of how the latter will be structured.

The auditor's suggestions on this section were important because they allowed us to improve the level of sub-section 2.4 "Economic analysis: monetary and contents", in which we integrate some elements that improved not only this subsection but also the general structure of the paper.

Answer to the comment 2b

We have not integrated subsection 2.2 and 2.3, as they concern two different moments of our work, since if in the first we introduce LCA and the software used to support it, in the second we highlight the software used for Energy simulation i.e. software "Design Builder.

Comments 3:

3) in chapter four

4.3 LCA imaging

4.4. Economic analysis and Valuation Issues

→Combine 4.3 and 4.4 to 4.3(LCA)

Answer to the comment 3

We have changed the title of sub-section 4.3 as suggested by the reviewer.

Concerning the suggestion to include Sub-section 4.4 in 4.3, as already explained about the remark concerning the title, Sub-section 4.3 provides a thermo-technical and environmental items whose acknowledgement should be distinguished from the economic-financial one presented in sub-section 4.4, typically belonging to the field of the economic-financial sciences.

Comments 4:

4) in Chapter 5

Chapter 5 also comprehensively covered three topics of energy, environment and economy, which need to be reduced.

In other words, focus on life cycle assessments for reducing carbon emissions in public buildings. On the other hand, I would like to reduce the economic and environmental contents.

Therefore, if the above is supplemented, your paper is for the Carbon Footprint Reduction of Public Buildings and is expected to be a good topic and good content.

Answer to the comment 4

In reality, the document aims to integrate environmental issues into energy policies relating to public buildings, whose owners are now required to comply with European regulations implemented in Italy by various measures that offer specific incentives.

Consequently, it seemed useful to point out the relevance of the three topics that can hardly be reduced without misleading the reader.

Reviewer 2 Report

Dear authors,

Firstly I hope you are safe and wish you and your country all the best!

Thank you for interesting paper, I enjoyed reading it. I have only one concern regarding results. You stated An air change rate of 0.5 vol·h−1. Is this before or after the retrofit interventions? I didn't find it in paper. Those values should be different before and after since you changing the windows. Please give your opinion on this! Thank you and best regards,

Author Response

We would like to thank the reviewers and Editor for their detailed comments and suggestions for the manuscript.

We believe that the comments have identified important areas which required further analyses and improvement.

After completion of the suggested edits, the revised manuscript has benefitted from an improvement in the overall presentation and clarity.

Below, you will find a point by point description of how each comment was addressed in the manuscript. Original reviewer comments in black color and responses in red color.

Reviewer 2

Firstly, I hope you are safe and wish you and your country all the best!

Comments 1:

Thank you for interesting paper, I enjoyed reading it. I have only one concern regarding results. You stated an air change rate of 0.5 vol·h−1. Is this before or after the retrofit interventions? I didn't find it in paper. Those values should be different before and after since you changing the windows. Please give your opinion on this! Thank you and best regards.

Answer to the comment 1

Thanks for your comment and nice words. The simulations were carried out with 0,5 vol h-1 before and 0,3 vol h-1 after the intervention in order to consider the reduction of air leakage and the improvement of airtightness due to the replacement of the old windows.

Reviewer 3 Report

The research is deeply developed and supported by a reliable methodology. The relevancy of the subject is nowadays unquestionable to ensure future EU objectives in energy consumption and GHG emissions. The results, even based on one specific sample, show universal integrity for further exploration and tools developed to support decision-makers on the potential of renovation outcomes.

Comments:

1. "Table 4. Building's carbon footprint for a building life of 100 year" (100 years), it also assumes a 100-year lifespan for each solution? But the window frames, moving parts, last lower than that, depending on the region and use, from 30 to 40 years. How the research team accessed and assessed this information?

2. "Figure 4. List of the works for the implementation of the three energy-environmental retrofit actions."

2.1. Please, insert the source for those solutions' maintenance percentages. Because, other sources state very different numbers: non-intensive green roof, around 3,150% per year (on the first ten years) on to maintain the solution's workability, plus gardeners, logistics, tools, new species (replacement), fertilizers, irrigation, et cetera; slightly above, windows around 0,876% per year (on the first ten years); please, also, reconsider the thermal solution. Source: http://www.cype.it.

2.2. The new solutions' maintenance percentages should consider the discount of the original solutions.

Author Response

We would like to thank the reviewers and Editor for their detailed comments and suggestions for the manuscript.

We believe that the comments have identified important areas which required further analyses and improvement.

After completion of the suggested edits, the revised manuscript has benefitted from an improvement in the overall presentation and clarity.

Below, you will find a point by point description of how each comment was addressed in the manuscript. Original reviewer comments in black color and responses in red color.

Reviewer 3                                                                        

The research is deeply developed and supported by a reliable methodology. The relevancy of the subject is nowadays unquestionable to ensure future EU objectives in energy consumption and GHG emissions. The results, even based on one specific sample, show universal integrity for further exploration and tools developed to support decision-makers on the potential of renovation outcomes.

Comments 1:

"Table 4. Building's carbon footprint for a building life of 100 year" (100 years), it also assumes a 100-year lifespan for each solution? But the window frames, moving parts, last lower than that, depending on the region and use, from 30 to 40 years. How the research team accessed and assessed this information?

Answer to the comment 1

Thanks for your comment. The carbon footprint for a building life of 100 years was calculated, as reported in the methodology paragraph, with the tool LCA calculator based on Eco-invent database which builds on the method of life cycle assessment (LCA) as standardised by International Organisation for Standardisation (ISO) 2006. The tool allows evaluating each phase chronologically from the extraction and processing of raw materials, to the manufacturing, packaging and marketing processes, to the use, re-use and maintenance of the product, and on to its eventual recycling or disposal as waste.

Comments 2:

  1. "Figure 4. List of the works for the implementation of the three energy-environmental retrofit actions."

Answer to the comment 2

The caption has been changed:

Figure 4. List of works and items for the implementation of the three energy-environmental retrofit actions.

Comments 3:

2.1. Please, insert the source for those solutions' maintenance percentages. Because, other sources state very different numbers: non-intensive green roof, around 3,150% per year (on the first ten years) on to maintain the solution's workability, plus gardeners, logistics, tools, new species (replacement), fertilizers, irrigation, et cetera; slightly above, windows around 0,876% per year (on the first ten years); please, also, reconsider the thermal solution. Source: http://www.cype.it.

Answer to the comment 3

Thanks for your comment. The maintenance values are based on ISO ISO 15686-5:2017Buildings and constructed assets, Service life planning, Part 5: Life-cycle costing, ASHRAE Handbbok HVAC Applications 2011, Cap. 37 Owning and Operating Costs, and Cap. 39 Operation and Maintenance Management, ASHRAE, Atlanta, GA and UNI EN 15459:2008, “Prestazione energetica degli edifici - Procedura di valutazione economica dei sistemi energetici degli edifici”

Line 214-215

The calculations have been consequently updated (check Figure 4)

Comment 4

2.2. The new solutions' maintenance percentages should consider the discount of the original solutions.

Answer to the comment 4

Thank you very much for this comment in response of which we significantly integrated the method (lines 274-295) set-up of the model (see Figure 5) and the results (Table 8), as well as the related discussions (lines 584-588), Figure 6 and 7.

Round 2

Reviewer 2 Report

No comments, thank you.